# Equilibrium Adsorption of Organic Compounds (Nitrobenzene Derivative, Synthetic Pesticide, Dye, and Surfactant) on Activated Carbon from Single- and Multi-Component Systems

**DOI:** 10.3390/molecules30010088

**Published:** 2024-12-29

**Authors:** Magdalena Blachnio, Malgorzata Zienkiewicz-Strzalka, Anna Derylo-Marczewska

**Affiliations:** Department of Physical Chemistry, Institute of Chemical Sciences, Faculty of Chemistry, Maria Curie-Sklodowska University, M. Curie-Sklodowska Sq. 3, 20-031 Lublin, Poland; malgorzata.zienkiewicz-strzalka@mail.umcs.pl (M.Z.-S.); anna.derylo-marczewska@mail.umcs.pl (A.D.-M.)

**Keywords:** 4-chlorophenoxyacetic acid, 4-nitroaniline, surfactant, dye, adsorption, multi-component system, activated carbon

## Abstract

This work aimed to investigate the adsorption of organic compounds (4-nitroaniline and 4-chlorophenoxyacetic acid) on activated carbon in the presence of selected dyes (uranine and Acid Red 88) and surfactants (sodium dodecyl sulfate and hexadecyltrimethylammonium bromide). The adsorbent, i.e., the activated carbon RIAA (experimental activated carbon, Norit NV, Amersfoort, The Netherlands), was fully characterised by nitrogen adsorption/desorption isotherms, potentiometric titration, small-angle X-ray scattering, X-ray photoelectron spectroscopy, and transmission electron microscopy. The measurements of the adsorption isotherms of individual adsorbates from aqueous solutions were performed, and the Generalised Freundlich equation (GF) was used for their analysis. The influence of the properties of the co-adsorbates and the adsorbent on the efficiency of removal of 4-nitroaniline and 4-chlorophenoxyacetic acid from the water phase was discussed. A sieving effect was found—large dye and surfactant molecules do not penetrate the micropores but only locate at their entrances, limiting the availability of carbon adsorption space for the nitro compound and the pesticide. A very clear influence of the co-adsorbates’ concentration on the adsorption efficiency of the organic pollutants from the mixture was also observed. When the CMC (critical micelle concentration) value was exceeded in the system with surfactant as the co-adsorbent, a decrease in the competition effect on the adsorption of 4-chlorophenoxyacetic acid was observed. This is due to the formation of large aggregates of the surfactant in the solution, which are poorly adsorbed on the activated carbon.

## 1. Introduction

The presence of organic substances in the environment is undesirable because they can impair the functional properties of water—its colour, taste, and smell [1,2]. Chemical pollutants that pose a threat to human and animal health or life are particularly dangerous when they enter the aquatic environment through industrial production or human activities and remain in the environment, affecting living organisms in the long term [3,4,5]. Organic compounds also disturb the biological balance and the self-purification process of water. The most dangerous organic pollutants include pesticides [6,7,8], surfactants [9,10], nitrobenzene compounds [11,12,13], dyes [14,15], phenols [16,17], polycyclic aromatic hydrocarbons (PAHs) [18], oil contaminants [19,20], and humic substances [21,22,23].

Depending on their place of origin, organic substances that are treated as environmental hazards are defined as primary or secondary species [24]. The first group concerns pollutants occurring in natural waters, while the second group concerns compounds resulting from water and wastewater treatment processes (some oxidation products of the treatment of organic pollutants) [25]. In addition to natural sources, there is also a large group of anthropogenic sources, which include waste, domestic, and industrial effluents. They are the by-products of industrial processes and the result of human activities [26,27]. Nitrobenzene derivatives such as 4-nitroaniline are used as a raw material in many industries, including the synthesis of dyes [28], pharmaceutical products [29], rubber inhibitors [30], antioxidants, poultry medicines, and corrosion inhibitors [31], which are regularly released into the environment and affect living organisms [32], requiring effective removal [33,34]. Dyes are mainly used in the textile industry [35] but they are also used in the manufacture of paper products and pharmaceuticals. A significant proportion of dyes and their degradation products are toxic [36]. In addition, dyes contained in water absorb light and impair photosensitivity. Azo dyes (e.g., Acid Red 88) are the largest group of dyes used in the textile industry and account for 60–70% of all dyes produced. Uranine is perfectly miscible with water and serves as a leak detector or tracer in groundwater due to its intense fluorescent colour [37,38]. Dyes can be degraded under anaerobic conditions to more dangerous intermediates and are responsible for the intense colouration of the areas they pollute. Due to the mass use of dyes, it is necessary to work on ways to remove them from the environment [39,40,41,42]. Finally, among anthropogenic pollutants (such as 4-CPA), pesticides with teratogenic, mutagenic, carcinogenic, and embryotoxic properties are particularly monitored and need to be removed from the aquatic environment [43]. The effects of the listed substances depend on their solubility in water and the form in which they occur. Some potentially insoluble compounds may dissolve in water in the presence of other compounds that act as solvents (e.g., polycyclic aromatic hydrocarbons) [44,45,46].

The adsorption process is used to effectively remove dissolved organic compounds from water and is an important element of many systems used in water treatment technology [47,48,49]. Sorption is usually one of the last elements of the system and should precede processes that remove suspensions and colloidal substances from the purified media. The adsorption process on activated carbon is very effective in removing organic substances, by-products of chemical oxidation and disinfection, and also allows the removal of certain amounts of viruses and some inorganic pollutants [50,51,52]. The effectiveness and usefulness of activated carbon in water treatment is determined by the adsorption capacity, the specific surface area, the pore size and its distribution, the chemical nature of the surface, and the particle size [53]. The most commonly used material for the adsorption of pollutants is activated carbon in granular or pelletised form. The use of granular activated carbon in water treatment is currently considered the best available method for removing organic compounds, especially those present in low concentrations [54,55]. The adsorption efficiency of the discussed compounds on activated carbon can be compared with other types of sorbents. In most cases, activated carbon is more effective in the adsorption of 4-nitroaniline [56], 4-CPA [57,58], Acid Red 88 [59], and uranine [60] than silica, polymeric, and other inorganic sorbents [61,62,63,64,65]. The high effectiveness of activated carbon in removing PAHs from water has been repeatedly confirmed. The PAH removal efficiency ranged from several dozen % to even 90–100% [66,67]. Acenaphthene, phenanthrene, fluorene, and anthracene can be removed with 50%, 85%, 88.3%, and 91% efficiency [68,69]. The removal efficiency of anthracene and dibenzopyrene on activated carbons can be very high. It was found that almost complete removal of anthracene was achieved after a contact time of 20 h. Some results show that activated carbons produced from agricultural waste are a very good adsorbent for compounds from the PAH group [70,71]. The complete removal of pesticides from aqueous solutions is not easy but purification with activated carbon facilitates an effective reduction in their content. Recently, the synthesis of carbon composites containing various metals has been proposed to preconcentrate or extract organic contaminants from aqueous solutions [72,73,74].

The monitoring and elimination of residues of pesticides, pharmaceuticals, and other harmful or toxic commercial chemical compounds in the environment is currently one of the fundamental tasks of sustainable development and green chemistry programs.

This paper presents the results of an investigation on the adsorption of organic pollutants, e.g., 4-nitroaniline and 4-chlorophenoxyacetic acid, on the activated carbon RIAA in the presence of selected dyes (uranine and Acid Red 88) and surfactants (sodium dodecyl sulfate and hexadecyltrimethylammonium bromide). The structural, textural, morphological, and physicochemical features of the adsorbent were studied utilising various complementary techniques. The influence of the properties of the co-adsorbates and the adsorbent on the efficiency of removal of 4-nitroaniline and 4-chlorophenoxyacetic acid from the water phase is discussed.

## 2. Results and Discussion

### 2.1. Characterisation of the Material

Figure 1 shows the nitrogen adsorption–desorption isotherm for the activated carbon RIAA, which falls under type I with an H4 hysteresis loop according to the IUPAC classification [75]. The shape of the experimental isotherm suggests that the carbon material has a mixed porosity profile and is rich in micropores with a notable amount of mesopores as well. The micropores are filled at low relative pressures, and at higher relative pressures, multilayer adsorption is observed, followed by capillary condensation occurring within the mesopores, highlighting their role in the adsorption process. Capillary condensation within these pores initiates at a relative pressure (p/p_0_) above 0.45, indicating a substantial presence of narrow slit-like mesopores. The BET surface area (S_BET_) and pore volume for RIAA are 1468 m^2^/g and 0.80 cm^3^/g, respectively (Table 1). The BJH pore size distribution, with Faas correction, displayed in Figure 1B, indicates that the carbon has pore diameters in the range of 3.5 nm to 4.5 nm. For micropore size distribution calculations from the nitrogen adsorption isotherm, both the Horvath–Kawazoe (HK, using slit pore geometry) and nonlocal density functional theory (NLDFT) methods were applied.

According to the HK model, the average pore size of RIAA is 0.75 nm, with a distribution up to 5 nm (Figure 1C). The proportion of micropores to mesopores in RIAA was found to be 40% and 60%, respectively. The NLDFT data calculated from model, N_2_ at 77 K on carbon slit pores, further confirms the distribution of slit-like pores within a range of 1 to 5 nm (Figure 1C).

The pH value at the point of zero charge (pH_PZC_) was measured for the activated carbon RIAA to evaluate the acid–base properties affecting the mechanism of the adsorption process. The pH_PZC_ value of activated carbon is a crucial factor because anion adsorption is favoured when the pH of the solution is below pH_PZC_, resulting in a positively charged surface. Conversely, when the solution pH is above pH_PZC_, the surface charge of the porous material becomes negative. Figure 2 shows that the pH_PZC_ value of the carbon is 9.2, meaning that the material maintains a positive charge under the experimental conditions. The alkalinity of activated carbon is probably related to the presence of groups such as diketones (quinones)—with an atomic content of 11.6%—as well as carbonyl groups (49.2 At %), compared to acid groups like phenol groups (33.2 At %). The presence and significant content of these groups were confirmed in XPS tests (Figure 3).

The high pH_PZC_ value indicates the presence of a large number of basic functional groups. In order to characterise and identify all of the functional groups present on the adsorbent surface, an analysis of the surface elemental composition was performed. The overall chemical composition of the activated carbon RIAA was precisely characterised by XPS (X-ray photoelectron spectroscopy). A representative survey spectrum of the carbon material and the extracted XPS atomic concentrations of the major elements are shown in Figure 3A, with a standard deviation of less than 0.2 atomic % for all elements. The XPS overview spectrum shows the presence of carbon (C1s), oxygen (O1s), and trace amounts or at the detection limit of nitrogen (N1s), silicon (Si2p), and chlorine (Cl2p) as impurities in the activated carbon. The analysis of the C1s core level indicates the aromatic character of the material according to the sp^2^-hybridised network. Figure 3B lists six components of the C1s core level, such as C=C (284.5 eV, 69.9 at. %), C–C/C-H (284.9 eV, 12.3 at. %), C–OH (286.5 eV, 7 at. %), C=O (287.7 eV, 2 at. %), COOH/COOR (289.0 eV, 3.5 at. %), and π-π* interactions (290.8 eV, 5.2 at. %). O1s scans (Figure 3C) show four main peaks at about 530.6 eV and 532.3 eV, 533.7 eV, and 535.0 eV, indicating the presence of quinine, carboxyl groups, aliphatic groups, and water, respectively. These oxygen-containing functional groups on the carbon influence the polarity, the acid–base properties of its surface, and may be involved in the adsorption of target compounds.

The sp^2^-hybridised network of RIAA activated carbon with the highest contribution to C1s signal (~70 At%) is shown in Figure 4. Figure 4 also illustrates the remaining functional groups on the surface, divided into those with acidic and basic properties. In the case of RIAA activated carbon, the basic functional groups predominate.

A high-resolution transmission electron microscopy study has been performed to estimate structure, porosity, and homogeneity/heterogeneity of the activated carbon RIAA. The TEM images (Figure 5) show a disordered hierarchical porous structure with regions showing partially ordered graphene layers. This indicates the presence of amorphous nanostructures with mixed porosity. TEM images show regions with tightly corrugated individual carbon layers with mesopores as well as parallel graphene sheets with slit-like pores. Such a microstructure creates porous spaces within the activated carbon material and enhances its ability to adsorb organic molecules. In addition, TEM images from different areas of the sample show a similar structure throughout the carbon material, a property that is particularly valuable for the intended applications.

An experimental SAXS curve was used to evaluate the structural features of the discussed carbon material. The experimental SAXS pattern (Figure 6A) shows features of a continuous curve that is unbroken over the entire angular measurement range. This type of activated carbon is characterised by a random distribution of scattering heterogeneities (pores) and the lack of correlation of their relative positions (absence of significant signals and scattering gains). The porous carbon RIAA shows a monotonically decreasing course of the scattering curve over the entire angular measurement range. This indicates a random distribution of scattering heterogeneities in the carbon structure. SAXS data plotted in double logarithmic coordinates (log–log) show a linear relationship with a power-law exponent of α = 3 (Figure 6B). This result indicates the fractal distribution of the observed heterogeneities. The value of α = 3.0 for RIAA could indicate the presence of small-scale fractal structures generated by carbon nanoclusters. The size of the fractal structure can be calculated using the formula L ≈ 2π/q^2^ and corresponds to L~4 nm for RIAA carbon. The volume-weighted particle size distribution Dv(R) (Figure 6C) shows the presence of a large number of objects with dimensions below 2 nm. The function p(r) allows for the refinement of the structural information by analysing the histogram of distances between pairs of points within the scattering object. The value 2Rs is the maximum diameter in the spherical particle (for the spherical calculation model), and Hc is the length of the cylindrical object with a diameter of 2Rc for the model describing the cross-section for monodisperse rod-shaped particles (Figure 6D).

### 2.2. Equilibrium Adsorption from Single-Component Systems

Figure 7 shows the adsorption isotherms for 4-nitroaniline (4-NA), 4-chlorophenoxyacetic acid (4-CPA), Acid Red 88 (AR 88), and uranine (U) from single-component solutions on the activated carbon RIAA. Considering the fact that the adsorbent used is characterised by a wide variety of pore sizes, including both micropores and mesopores, the experimental data obtained were analysed based on the theory of adsorption on energetically inhomogeneous solids. According to the assumptions of this theory, the adsorption from dilute solutions can be represented by the global integral Equation (1). The concept of dilute solutions refers to the systems in which the excess adsorption of the solute is approximately equal to its adsorbed amount in the solid surface phase.
(1)$$\theta =\frac{a}{a_{m}}=\int_{∆E}\theta_{1}\left(c,E\right)\chi \left(E\right)dE$$
where θ—relative adsorption of the organic compound in the surface phase; a—adsorption at a given equilibrium concentration; a_m_—adsorption capacity; θ_1_—local relative adsorption; c—solution concentration; E—difference between reduced adsorption energies of the dissolved compound and the solvent treated as a reference component with a constant concentration; χ(E)—function of the distribution of adsorption energy.

The above integral equation generates a series of isotherms that can be divided into two groups, namely equations that are special forms of the Generalised Langmuir equation (GL), and the Dubinin–Astakhov equation (D-A). In this work, the GL equation was used to analyse the experimental data [76,77,78,79]:(2)$$\theta ={\left[\frac{{\left(Kc_{eq}\right)}^{n}}{1+{\left(Kc_{eq}\right)}^{n}}\right]}^{m/n}$$
where m and n—heterogeneity parameters characterising the shape (width and asymmetry) of the adsorption energy distribution function, having values from the range (0,1); K—equilibrium constant describing the position of the distribution function on the energy axis.

For specific values of heterogeneity parameters, the Generalised Langmuir equation reduces to known adsorption isotherm equations corresponding to different symmetries of adsorption energy distribution:The Langmuir–Freundlich equation (LF), when m = n ϵ (0,1):
(3)$$\theta =\frac{{\left(Kc_{eq}\right)}^{m}}{1+{\left(Kc_{eq}\right)}^{m}}$$The Generalised Freundlich equation (GF), when n = 1 and m ϵ (0,1):
(4)$$\theta ={\left[\frac{Kc_{eq}}{1+Kc_{eq}}\right]}^{m}$$The Tóth equation (T), when m = 1 and n ϵ (0,1):
(5)$$\theta =\frac{Kc_{eq}}{{\left[1+{\left(Kc_{eq}\right)}^{n}\right]}^{1/n}}$$The Langmuir equation (L), when m = n = 1:
(6)$$\theta =\frac{Kc_{eq}}{1+Kc_{eq}}$$The Classical Freundlich equation (CF), when m ϵ (0,1):
(7)$$\theta ={\left(Kc_{eq}\right)}^{m}$$

The selection of the most optimal form of the GL equation from Equations (3)–(7) was based on the method of minimising the sum of squared deviations of the adsorption values and applying a limit to both the adsorption capacity a_m_ and the equilibrium constant K. It was found that the Generalised Freundlich equation (GF) is the most appropriate to describe all four adsorption systems, and its parameters are used in the further discussion (Table 2). The GF equation corresponds to a strongly asymmetric distribution of the adsorption energy (extended towards high energies). The agreement of the theoretical isotherms with the experimental data confirms the high quality of the selected fit (Figure 7A).

Analysing the obtained experimental data, one can say that the adsorption efficiency of organic substances on RIAA is differentiated, and the greatest values occurred for 4-NA (a_m_ = 4.55 mmol/g), slightly worse for 4-CPA (a_m_ = 3.59 mmol/g), and the worst for AR 88 (a_m_ = 1.09 mmol/g) and U (a_m_ = 0.59 mmol/g). This is largely due to the differences in the physicochemical properties and molecular structure of the adsorbates. Let us, therefore, investigate the influence of the following parameters and molecular features of adsorbates on the adsorption capacity: (i) hydrophobic–hydrophilic properties, (ii) molecular state, (iii) type of functional groups, and (iv) size and spatial shape. The values of some physicochemical properties and possible molecular forms of the studied pollutants are presented in Appendix A and Appendix A.

The measure of the hydrophobicity of a substance is its solubility in water (a lower solubility indicates a greater hydrophobicity of the adsorbate). According to the principle of similarity, the greater the hydrophobicity of a substance, the stronger its affinity to a solid of a similar chemical nature. In such a case, an adsorption process can be based on the mechanism of hydrophobic interactions. Due to the fact that the activated carbon RIAA is characterised by a high value of the zero charge point, pH_pzc_ = 9.2—which indicates a strong hydrophobicity of its surface—the hydrophobicity of the adsorbate should be the driving force in the adsorption process. The results of the adsorption measurements correlate well with the above assumption (increase in adsorption with decreasing solubility of the substance). However, the presentation of the experimental adsorption isotherms in the coordinate system reduced by the solubility parameter of the adsorbates (lack of overlap of the experimental data) indicates a considerable influence of additional factors on the efficiency of adsorption.

Let us analyse the molecular state of the adsorbates and the surface charge of the adsorbent under the experimental conditions (pH~7). According to the distributions of the chemical forms of the adsorbates in Appendix A, 4-nitroaniline appears in the solution in a neutral form, while the other pollutants are present in an anionic form. As can be seen from the potentiometric titration measurements (Figure 2), the surface of RIAA has a positive electrical charge under the considered pH conditions. Such surface characteristics of the activated carbon result from the significant proportion of basic oxygen groups with a structure corresponding to chrome or pyrone-like structures compared to acidic complexes (XPS measurement results, Figure 3). The molecular state of the adsorbate and the surface chemistry of the activated carbon determine the mechanism and strength of adsorption. Electrically neutral molecules (4-NA) are bound based on dispersive interactions between the π-electrons of their aromatic ring and the graphene planes of the carbon. Anionic forms of pollutants (4-CPA, AR 88, and U) in turn, are adsorbed via electrostatic interactions with basic centres on the carbon surface. The dispersion interactions exceed the latter mechanism, which is consistent with the research results obtained (a_m_ for 4-NA > a_m_ for the other adsorbates).

The influence of functional groups on the aromatic rings of the tested substances on their adsorption mechanism on RIAA can be explained. This is not directly related to the interaction of the functional groups with the adsorbent but to changes in the molecular properties of the adsorbed compound. Depending on their nature, they can attract or repel electrons and, thus, influence the dispersion interactions in the adsorbate–adsorbent system. Functional groups that are electron donors activate the aromatic ring by moving electrons towards it, thus increasing the interactions of the adsorbate molecule with the π-electrons of the graphene planes of the carbon. Functional groups of this type include amine (–NH_2_), hydroxyl (–OH), and carbonyl (=O), which are present in 4-NA, AR 88, and U molecules. On the other hand, deactivating groups, which are electron acceptors, lead to a decrease in the electron density in the aromatic ring and, thus, to a weakening of the interactions with the carbon surface. This type is represented by the groups nitro (–NO_2_), chloride (–Cl), sulfonate (–SO_3_H), and carboxyl (–COOH), which are present in the molecules of all aromatic compounds tested. It is likely that the presence of electron-donating and electron-accepting groups in the adsorbate molecules and on the surface of the activated carbon allows the formation of donor–acceptor complexes, which are to some extent responsible for the adsorption process. In turn, the presence of electronegative atoms, i.e., nitrogen, oxygen, sulfur, and chlorine in the adsorbate molecules suggests the possibility of the formation of hydrogen bonds with the surface groups of the activated carbon.

One of the important factors determining the adsorption efficiency is the size and spatial shape of the adsorbate molecules in relation to the pore size of the adsorbent. As can be seen from the parameters describing the geometry (van der Waals volume and maximum projection area; Appendix A), the tested substances are diverse in this respect, and the porous structure of the adsorbent includes both micro- and mesopores. Therefore, some of the pores may be inaccessible to large organic molecules, i.e., AR 88 and U, due to the sieving effect. This significantly limits the possibility of using the internal adsorption space for the dye adsorption process. The weaker adsorption of uranine compared to Acid Red 88, despite its relatively smaller molecular size, can be explained by hydrophilic properties. Polar groups in the uranine molecule can be surrounded by water molecules, which increases the actual molecule size (together with the hydration shell) and weakens the interactions in the adsorbate–adsorbent system.

To summarise, the adsorption of aromatic compounds on activated carbon is a complex process and depends on many parameters that affect both the pollutant dissolved in the water and the solid. The activated carbon RIAA is more suitable for the adsorption of neutral (4-NA) or non-polar substances (4-CPA) with a small molecular size due to its high hydrophobicity and high proportion of micropores, while it has a low adsorption capacity for large (AR 88 and U) and especially large and polar (U) pollutants. Using uranine as an example, one can observe a strong negative effect of both the chemical nature of the carbon surface and its porosity properties on the adsorption capacity. Moreover, the uranine/RIAA carbon adsorption system shows the lowest energetic heterogeneity (the highest value of the parameter m), which means that the affinity of uranine leading to its adsorption is limited to selected adsorption sites.

The adsorption efficiency of the discussed compounds on various types of sorbents is presented in Table 3.

To prove the precision and repeatability of experiments, the adsorption measurements for 4-NA and 4-CPA on the activated carbon RIAA were performed in three individual series. The adsorption data were optimised using the Generalized Freundlich equation (GF) along with a statistical analysis of the parameters a_m_, m, and K. The obtained results are graphically depicted in Appendix A. The lowest standard deviations were obtained for the average parameter a_m_ (±1.6% and ±3.7% for 4-NA and 4-CPA, respectively), while for the average heterogeneity parameter m and the average adsorption constant K, standard deviations were in the range of ±4.4–±11%. Such dispersions of the adsorption measurement results are acceptable and confirm their reliability and the precision of the applied methodology.

### 2.3. Equilibrium Adsorption from Multi-Component Systems

Since ideal single-component solutions are practically absent in the environment, it is necessary to investigate more comprehensive interdependencies between co-occurring adsorbates and their relationship with the adsorbent, which allows for expanding the knowledge of the mechanism of their adsorption on activated carbon. Two series of adsorption measurements of multi-component systems were carried out to evaluate the effect of accompanying substances on the adsorption efficiency of selected groups of compounds. The first series comprised the adsorption of (i) 4-nitroaniline in the presence of uranine (4-NA + U) and (ii) 4-nitroaniline in the presence of Acid Red 88 (4-NA + AR 88). The second series involved the adsorption of (i) 4-chlorophenoxyacetic acid in the presence of sodium dodecyl sulfate (4-CPA + SDS) and (ii) 4-chlorophenoxyacetic acid in the presence of hexadecyltrimethylammonium bromide (4-CPA + HTAB).

The equilibrium concentrations of the binary solutions were calculated based on Beer–Walter’s law and the additivity law. The latter law assumes that in a multi-component system, if several different components of the solution absorb radiation at a certain wavelength, the absorbance is equal to the sum of the absorbance for the individual components present in the system. The condition for the system to fulfil the additivity law is the absence of interactions between its components. Accordingly, the equilibrium concentrations of the components were calculated based on the absorbance values obtained for binary solutions and the absorbance coefficients determined for individual substances. Mathematically, the determination of the equilibrium concentrations of the components in the binary system consisted of solving the system of two equations with two unknowns:ABS(λ_1_) = c_A_ε_A1_ + c_B_ ε_B1_(8)
ABS(λ_2_) = c_A_ε_A2_ + c_B_ ε_B2_(9)
where c_A_, c_B_—the concentrations of substances A and B in a two-component solution; ABS(λ_1_), ABS(λ_2_)—the absorbance values determined at the wavelengths λ_1_ and λ_2_ for the spectrum measured for a two-component solution; ε_A1_, ε_A2_—the molar absorption coefficients of substance A at the wavelengths λ_1_ and λ_2_ determined for one-component solutions.

#### 2.3.1. Equilibrium Adsorption in Nitro–Compound–Dye Systems

In this subsection, the effect of two dyes on the adsorption of 4-nitroaniline was discussed to evaluate the influence of competing solutes on the effectiveness of the adsorption process. In Figure 8A, the experimental adsorption isotherms of 4-nitroaniline from solutions with uranine at different concentration ratios of both adsorbates in the initial solutions were compared with the adsorption isotherm of 4-NA from single-component solutions. A significant influence of the dye concentration on the adsorption of 4-nitroaniline was found. With decreasing uranine concentration in the solution, a distinct increase in 4-NA adsorption can be observed; the maximum 4-NA adsorption was reached at a concentration ratio of 3:1, and it was similar to the adsorption value for the single-solute system. This means that both adsorbates partially compete for the different adsorption sites of the activated carbon. The larger uranine molecules are adsorbed in larger pores (mesopores), while the smaller 4-nitroaniline molecules mainly fill micropores. We can, therefore, conclude that the adsorption of both compounds is partly independent of each other. At higher uranine concentrations in the mixture, the process of micropore blocking by its large molecule restricts the accessibility of this adsorption space for 4-NA, reducing the adsorption efficiency of 4-NA.

Figure 8B shows the adsorption isotherms of 4-nitroaniline from solutions with Acid Red 88 at different concentrations in relation to 4-NA adsorption from a single-component solution. Similarly, as in the case of systems with uranine, the adsorption of 4-nitroaniline decreases but the effect of AR 88 on 4-NA adsorption is stronger than that of uranine. The linear shape of the AR 88 molecules likely allows them to diffuse and become more densely trapped in smaller pores of the adsorbent than the spatially extended uranine molecules. In this case, we are dealing with stronger competitive adsorption between 4-NA and AR 88 than between 4-NA and U.

Figure 9A–C compares the adsorption of 4-nitroaniline in the presence of uranine and Acid Red 88 as competing substances (while maintaining the same concentration ratios of 4-NA to both dyes in the initial solutions, i.e., 3:1, 2:1, and 1:1) relative to the adsorption of 4-NA from single-component solutions. It can be clearly seen that for all systems the presence of Acid Red 88 in the adsorbate solutions has a stronger effect on the 4-NA adsorption. This dye significantly reduces the adsorption of 4-nitroaniline (Δa was lower by about 24%, 29%, and 41% at c_eq_ = 1.75 mmol/L). In the case of uranine, this effect is clear at its higher concentration in the solution (Δa was lower by about 20% and 25% at _ceq_ = 1.75 mmol/L for initial concentration ratios of 4-NA to U of 2:1 and 1:1, respectively), while at a significantly higher ratio, i.e., 3:1, the adsorption of 4-nitroaniline is practically at the level of adsorption from single-component solutions (Δa was lower by about 1% at c_eq_ = 1.75 mmol/L). These variations result from the differences in the structure, shape, and molecular size, as well as the physicochemical properties of the two dyes. Uranine as a hydrophilic substance (with high solubility in water) has a lower affinity for the hydrophobic surface of the carbon than Acid Red 88. The lower the affinity of the co-adsorbate, the fewer molecules diffuse from the volume solution to the solution–solid interface to finally occupy the adsorption centres. In addition, the hydrophilic properties of uranine cause its molecules to become hydrated, which significantly slows down the diffusion rate in the solution and weakens the interactions with the adsorbent. All this makes uranine a weaker competitor in adsorption from binary solutions than Acid Red 88. Nevertheless, the molecular size of both dyes suggests that their presence limits the availability of adsorption space for 4-nitroaniline, mainly by blocking the entrances to smaller pores (micropores).

Figure 9D shows the effect of the presence of 4-nitroaniline on the Acid Red 88 adsorption with respect to the adsorption of this dye from single-component solutions. After analysing the experimental data obtained, we can conclude that the adsorption of Acid Red 88 on the activated carbon RIAA is lower (in molar values) than the adsorption of 4-nitroaniline. However, at lower concentrations than 4-NA, adsorption is observed close to the maximum (from the single-component system), which reflects relatively higher adsorption forces and an effect of the molecular size. This explains the fact that the presence of 4-NA has a lower effect on the dye adsorption (Δa was lower by about 25% at c_eq_ = 1.75 mmol/L) than the presence of AR 88 on the 4-NA adsorption (Δa was lower by about 41% at c_eq_ = 1.75 mmol/L). The given percentage changes in the adsorption refer to a two-component system with an initial adsorbate concentration ratio of 1:1 relative to single-component systems.

#### 2.3.2. Equilibrium Adsorption in Pesticide–Surfactant Systems

This study investigated the effect of surfactants of different chemical nature on the adsorption of 4-chlorophenoxyacetic acid (4-CPA). For this purpose, long-chain compounds, i.e., sodium dodecyl sulfate (SDS) and hexadecyltrimethylammonium bromide (HTAB), of anionic and cationic nature, respectively, were used. The physicochemical properties of SDS and HTAB are as follows: molecular weight, 288.4 g/mol and 346.5 g/mol; water solubility, 200 g/L and 100 g/L; critical micelle concentration, 8.05 mmol/L and 0.92 mmol/L, respectively. Figure 10A,B shows the experimental adsorption isotherms of 4-CPA from a one-component solution and in the presence of surfactants (SDS or HTAB) on RIAA. For both groups of two-component systems, the adsorption process was carried out at the following pesticide-to-surfactant concentration ratios: 2:1, 1:1, and 1:2. In both groups, clear differences in the pesticide adsorption can be seen, which depend on the mutual ratios of the concentrations of the respective substances. As a rule, solutions with higher surfactant concentrations exhibit the lowest 4-CPA adsorption. A comparison of the geometric parameters of the substances shows that 4-CPA has the smallest molecule (the maximum projection area and van der Waals volume are 47.44 Å^2^ and 115.97 Å^3^, respectively). The values of these parameters for SDS and HTAB are 91.82 Å^2^ and 266.19 Å^3^ and 115.50 Å^2^ and 351.22 Å^3^, respectively. It follows that micropores and small mesopores are most suitable for the 4-CPA adsorption, while only mesopores are suitable for surfactants. As the surfactant concentration increases, the pore entrances may become blocked and the micropore space for the pesticide may be limited. They also reduce the adsorption efficiency of the pesticides by occupying adsorption sites in smaller mesopores. In the case of HTAB as a co-adsorbate, a certain deviation from this rule can be seen at twice the concentration of this substance. This may be due to exceeding the critical micelle concentration (CMC) for HTAB and the formation of much larger molecular associations, which are characterised by an absolute lower adsorption competitiveness against 4-CPA molecules. An increase in the concentration of 4-CPA in the solution in turn causes an increase in the adsorption of this compound in all cases.

The HTAB micelles formed in the solution most likely have a spherical structure, although the presence of 4-chlorophenoxyacetic acid can influence the change in their shape. As reported in the literature, spherical HTAB structures can transform into ellipsoidal, rod-shaped, and worm-like structures in the presence of different types of additives (e.g., phenols, aromatic amines, and aromatic acids) [80]. Regardless of the shape of the micelles, however, their core is formed by hydrocarbon chains interacting with each other through van der Waals forces, while the outer part is formed by positively charged hydrophilic groups. The size of the resulting micelles in combination with their net charge leads to a considerable reduction in the adsorption capability of the surfactant.

The difference in the efficiency of pesticide adsorption was also analysed in terms of the properties of a particular co-substance and the dependence of the adsorption on the ratio of the concentrations of 4-CPA and surfactant in the solution. For this purpose, the 4-CPA isotherms from binary solutions with the same concentrations but different co-adsorbates were compared (Figure 11A–C). From the isotherms in Figure 11A,B, it can be concluded that HTAB has the greatest effect on limiting the 4-CPA adsorption (Δa was lower by about 17% and 39% at c_eq_ = 2.5 mmol/L), although SDS also reduces pesticide adsorption (Δa was lower by about 11% and 21% at c_eq_ = 2.5 mmol/L). This could be due to differences in the structure and geometric size of the surfactant molecules. Larger HTAB molecules (the maximum projection area of 115.50 Å^2^) can more effectively prevent 4-CPA molecules from accessing active sites in micropores and small mesopores than SDS molecules (the maximum projection area of 91.82 Å^2^). In turn, comparing the isotherms at a concentration ratio of 1:2 (Figure 11C), it can be stated that the effect of both surfactants on the herbicide adsorption is comparable (Δa was lower by about 28% and 25% in the presence of SDA and HTAB, respectively, at c_eq_ = 2.5 mmol/L). This can be attributed to exceeding the critical micelle concentration (CMC) for HTAB, leading to the formation of difficult-to-adsorb molecular aggregates, which favours the increase in the 4-CPA adsorption.

The amphiphilic nature of surfactants means that their adsorption on activated carbon can occur via several mechanisms, i.e., hydrogen bonding, or electrostatic and hydrophobic interactions. However, considering the positive surface charge of the adsorbent and the cationic form of HTAB, it can be concluded that the contribution of hydrophobic interactions to the adsorption mechanism is dominant and their strength clearly exceeds the electrostatic repulsive force. Since the hydrophobic interactions in the surfactant-activated carbon system increase with the length of the alkyl moiety in the surfactant, the HTAB adsorption is higher than the SDS adsorption. This in turn also explains the greater effect of HTAB on the adsorption of 4-CPA from multi-component solutions compared to the effect of SDS as a co-adsorbent.

## 3. Materials and Methods

### 3.1. Materials and Chemicals

The activated carbon RIAA (Norit NV, Amersfoort, The Netherlands) was used as adsorbent after the ash removal process (treatment with hydrochloric acid at 60 °C for 6 h). The aromatic compounds 4-nitroaniline (4-NA) and 4-chlorophenoxyacetic acid (4-CPA) were used as adsorbates. In addition, Acid Red 88 (AR 88), uranine (U), sodium dodecyl sulfate (SDS), and hexadecyltrimethylammonium bromide (HTAB) were used as accompanying substances in a multi-component system; they were purchased from Sigma-Aldrich (St. Louis, MO, USA).

### 3.2. Methods and Calculations

The textural properties of the adsorbent were evaluated through low-temperature nitrogen adsorption–desorption isotherms measured at 77 K across the full range of relative pressures (0 to 950 mmHg) using a sorption analyser (ASAP 2020, Micromeritics, Norcross, GA, USA). The specific surface area (S_BET_) was determined from the experimental isotherms using the standard BET method. Pore size distribution curves were derived from the adsorption and desorption branches of the isotherm following the Barrett–Joyner–Halenda (BJH) model with cylindrical pores and Faas correction. Total pore volume (V_t_) was calculated based on the nitrogen adsorbed at p/p_0_ = 0.98, while micropore volume (V_mic_) was estimated using the t-plot method. Before analysis, the material sample was degassed at 90 °C and 1 mmHg for 24 h in the analyser’s degas port.

The acid–base characteristics of the carbon surface were evaluated using potentiometric titration. A suspension of the carbon, acidified and diluted with NaCl, was placed in a thermostatically controlled vessel at 25 °C and titrated with NaOH through an automatic burette (Dosimat 765, Metrohm, Herisau, Switzerland) connected to a pH meter (PHM240, Radiometer, Copenhagen, Denmark). By analysing the pH changes relative to the volume of titrant added, the surface charge density and the point of zero charge for the carbon were determined.

The chemical characterisation of the surface chemistry of the adsorbent, which provides information about the surface-active groups, the elemental composition, and the electronic state of the elements, was carried out using X-ray photoelectron spectroscopy (XPS). X-ray photoelectron spectroscopy data were collected on a Multi-chamber UHV System, Prevac (2009, Rogów, Poland), using the hemispherical analyser ScientaR4000 by monochromatic Al Ka radiation from a high-intensity source MX-650, Scienta (Uppsala, Sweden).

The textural properties of the activated carbon were analysed using the microscopic technique, including a high-resolution transmission electron microscope (HR-TEM), Titan G2 60–300 (FEI Company, Hillsboro, OR, USA), operating at an accelerating voltage of 200 kV.

The SAXS analysis was conducted using an Empyrean diffractometer (PANalytical, Malvern, UK) equipped with a Cu anode X-ray tube and the SAXS/WAXS sample stage in capillary mode. The device was powered by a 4 kW high-voltage X-ray generator with settings of 40 kV and 40 mA. The incident beam path utilised W/Si and a graded elliptical X-ray mirror. The SAXS setup operated with a 2θ range of −0.1 to 4 degrees, a step size of 0.005, and a counting time of 1.76 s, resulting in 821 points per scan. A Cu 0.2 mm beam attenuator was used near primary beam measurements. Data were collected via a PIXcel3D detector and a receiving slit of 0.05 mm active length. The scattering vector, q, was defined as q = 4πsinθ/λ, where 2θ is the scattering angle and λ is the X-ray wavelength (1.5418 Å). Background scattering was assessed using air-scattering measurements with an empty sample holder.

The adsorption isotherms of aromatic substances from single- and multi-component systems on the activated carbon RIAA were measured using the static method. In the experiment, 0.04 g samples of the adsorbent were placed in Erlenmeyer flasks with 5 mL of distilled water and degassed under vacuum. One- or two-component solutions were then added to the flasks. The prepared adsorption systems were shaken in the Innova 40 incubator at a constant temperature of 25 °C for 10 days. After reaching adsorption equilibrium, the absorbance spectra of the solution samples were measured using the Cary 4000 spectrophotometer (Varian Inc., Palo Alto, CA, USA). The adsorption measurements for 4-NA and 4-CPA on the activated carbon RIAA were performed in three individual series, repeating the entire experimental cycle each time. Adsorption from a multi-component system was carried out in two series of measurements. The first series comprised the adsorption of (i) 4-nitroaniline in the presence of uranine (4-NA + U), and (ii) 4-nitroaniline in the presence of Acid Red 88 (4-NA + AR 88). The molar ratios between the nitro compound and the accompanying substance in the initial solutions were 3:1, 2:1, and 1:1. The second series involved the adsorption of (i) 4-chlorophenoxyacetic acid in the presence of sodium dodecyl sulfate (4-CPA + SDS), and (ii) 4-chlorophenoxyacetic acid in the presence of hexadecyltrimethylammonium bromide (4-CPA + HTAB). The molar ratios of the pesticide to the accompanying substance in the initial solutions were 2:1, 1:1, and 1:2.

## 4. Conclusions

The adsorption of 4-nitroaniline (4-NA, a_m_ = 4.55 mmol/L) on the activated carbon RIAA is greater than the adsorption of two dyes, uranine (U, a_m_ = 0.59 mmol/L) and Acid Red 88 (AR 88, a_m_ = 1.09 mmol/L). This is due to the structure of their molecules, molecular state, and physicochemical properties. The 4-NA molecule is small compared to the dye molecules, occurs in solution in undissociated form, and is also characterised by poor solubility. This allows its molecules to penetrate into smaller pores and adsorb there according to the mechanism of dispersion, hydrophobic and hydrogen interactions, and the formation of donor–acceptor complexes. Dye molecules have a much larger size and a more developed structure. For this reason, they cannot penetrate into micropores but only adsorb in larger pores (mesopores). Furthermore, they occur in solution in dissociated form, which favours the electrostatic interactions. The adsorption of dyes based on the formation of hydrogen bonds and donor–acceptor complexes is also possible;In the case of the presence of uranine in the mixture (different concentration ratios of both adsorbates in the initial solution), a different effect of dye concentration on the adsorption of 4-nitroaniline was observed compared to the adsorption of 4-NA from a one-component solution (Δa was lower by about 1–25%). With decreasing uranine content in the mixture, the adsorption of 4-NA increases significantly and reaches an adsorption close to that of single-component solutions in a system with a concentration ratio of 4-NA to U of 3:1 (Δa was lower by about 1%). This could mean that the adsorption of both compounds is partially independent of each other. Larger uranine molecules adsorb in larger pores, while small 4-nitroaniline molecules fill the micropores. At higher contents of uranine in the mixture, some of the micropores are blocked by its large molecules, which reduces the adsorption of 4-NA;In the case of the presence of Acid Red 88 in the mixture (different concentration ratio of both adsorbates), a stronger effect of this dye on the adsorption of 4-nitroaniline (Δa was lower by about 24–41%) was observed than for uranine. For the system with an equal content of 4-NA and AR 88 (1:1), the adsorption of 4-nitroaniline decreases significantly in the presence of Acid Red 88 (Δa was lower by about 41%). The higher the concentration of 4-nitroaniline in the mixture, the lower the decrease in 4-NA adsorption (Δa was lower by about 24%). Due to the greater affinity of Acid Red 88 for the hydrophobic surface of carbon compared to uranine, it more effectively blocks the entrances to smaller pores (micropores), thereby limiting the availability of adsorption space for 4-nitroaniline to a greater extent. The adsorption of Acid Red 88 on the activated carbon is lower (in terms of number of moles) than the adsorption of 4-nitroaniline but adsorption near the maximum is observed at lower concentrations than for 4-NA (from the single-component system). This means that relatively higher adsorption forces and an effect of the molecular size occur. Therefore, the presence of 4-NA in the solution has less influence on the dye adsorption (Δa was lower by about 25%) than the presence of AR 88 on the adsorption of 4-NA (Δa was lower by about 41%);The adsorption capacity a_m_ of 4-chlorophenoxyacetic acid (4-CPA) on the activated carbon RIAA is 3.59 mmol/L. In the presence of sodium dodecyl sulfate (SDS) and hexadecyltrimethylammonium bromide (HTAB), a decrease in the adsorption of 4-CPA was observed, which was proportional to the increase in its concentration in the initial solution. This could be due to the large size and shape of the surfactants mentioned, which make it difficult for the pesticide molecules to reach the active sites in the micropores. The adsorption of pesticides on activated carbon is stronger than that of surfactants and proceeds according to the mechanism of electrostatic, hydrophobic, and hydrogen interactions, and the formation of donor–acceptor complexes. The adsorption of surfactants is mainly controlled by hydrophobic interactions and to a lesser extent by hydrogen and electrostatic bonds;Below the CMC point for HTAB, a greater influence on the adsorption of 4-CPA (Δa was lower by about 17% and 39%) is observed compared to SDS (Δa was lower by about 11%, 21%, and 28%). This is probably due to the larger dimensions of its molecules, which decrease the accessibility of the internal space of the activated carbon for 4-CPA. When the CMC value is exceeded, a decrease in the influence of HTAB on the adsorption of 4-chlorophenoxyacetic acid is observed (Δa was lower by about 25%). This is due to the formation of large aggregates of the surfactant in the solution, which are poorly adsorbed on the activated carbon;Increasing the concentration of 4-CPA in the binary solution reduced the difference between its adsorption from one- and two-component solutions in every case studied.

## Figures and Tables

**Figure 1 molecules-30-00088-f001:**
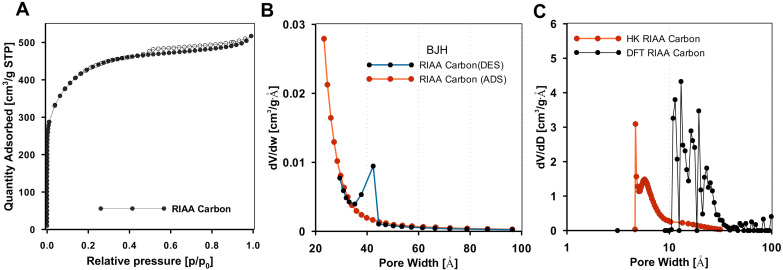
(**A**) Nitrogen adsorption–desorption isotherm of the activated carbon RIAA ((Full sphere marks indicate adsorption, hollow spheres indicate desorption)) and, (**B**) BJH pore size distribution curves, (**C**) micropore size distributions determined as the Horvath–Kawazoe differential pore volume plot and nonlocal density functional theory (NLDFT).

**Figure 2 molecules-30-00088-f002:**
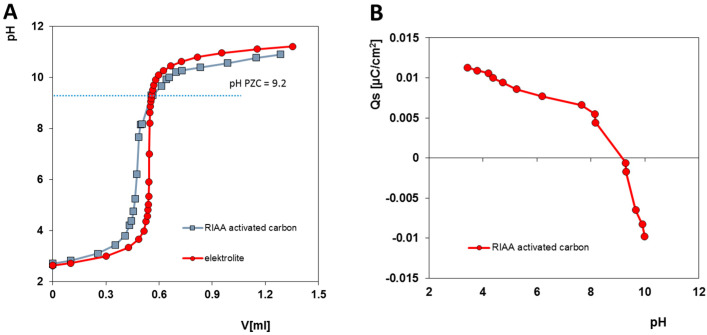
(**A**) Potentiometric titration curves for an RIAA suspension and blank NaCl electrolyte. (**B**) Dependences of surface charge density of the activated carbon RIAA on solution pH.

**Figure 3 molecules-30-00088-f003:**
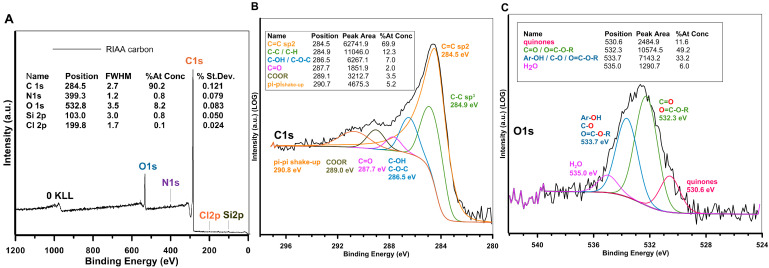
(**A**) Survey scan XPS spectrum of the carbon RIAA and details of elemental analysis. (**B**) High-resolution core-level spectra from the O1s region and (**C**) high-resolution core-level spectra from the C1s region with identification and contents of individual core levels.

**Figure 4 molecules-30-00088-f004:**
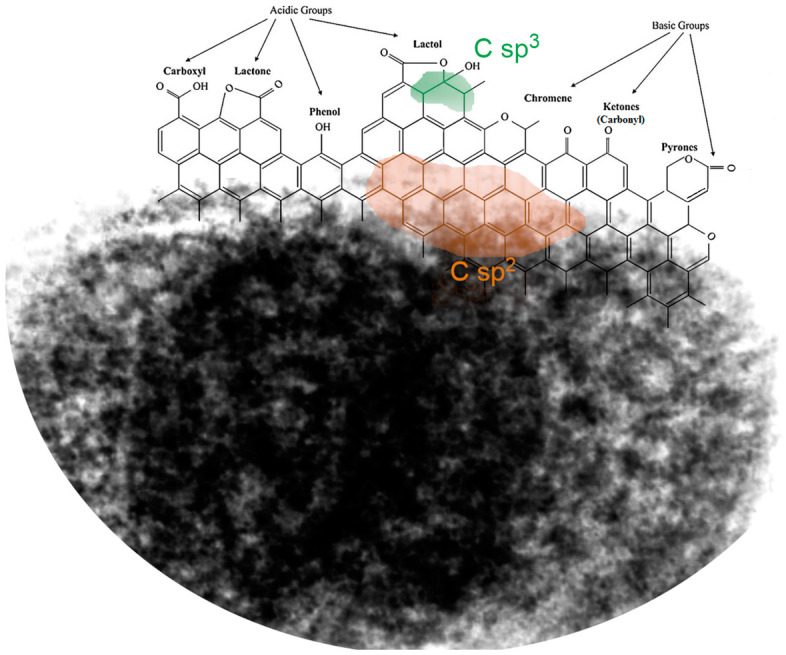
Activated carbon surface model with functional groups of various nature.

**Figure 5 molecules-30-00088-f005:**
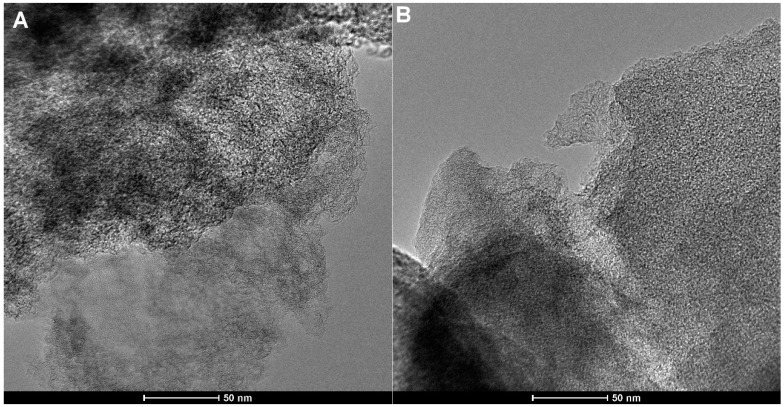
Transmission electron micrograph (TEM) of the activated carbon RIAA for two sample areas at lower (**A**,**B**) and higher magnification (**C**,**D**), enlargement of the image with visible graphite layers as an insert in image (**D**).

**Figure 6 molecules-30-00088-f006:**
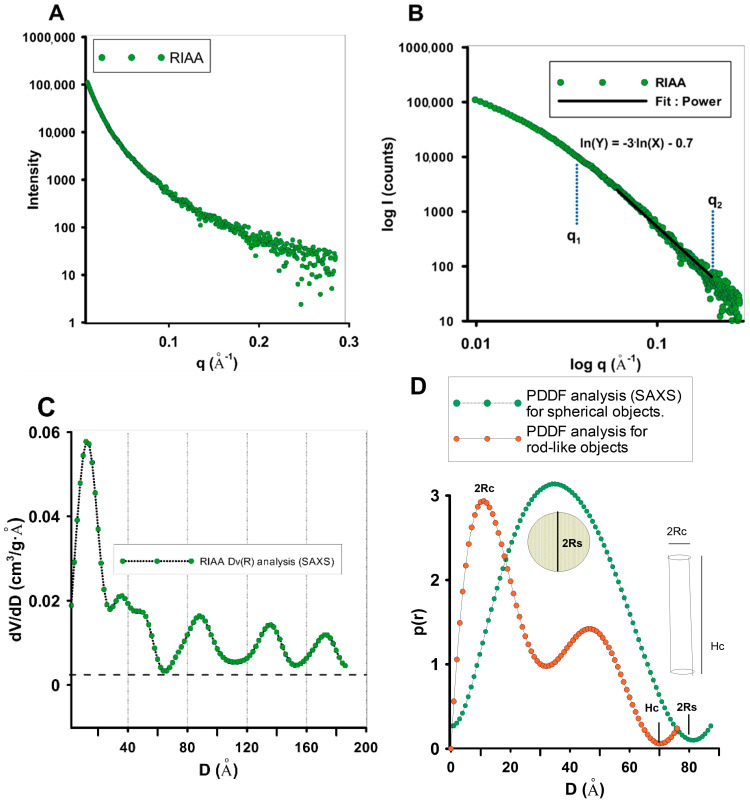
Small-angle scattering from the activated carbon (**A**). Representative SAXS curve in range of scattering vectors to 0.3 Å^−1^ (**B**). The log–log plot of SAXS intensity in the power-law range (**C**). Volume-weighted particle size distribution Dv(R) from the scattering curve of an ensemble of spherical particles and pair distance distribution function p(r) for globular and rod-type models of particles (**D**).

**Figure 7 molecules-30-00088-f007:**
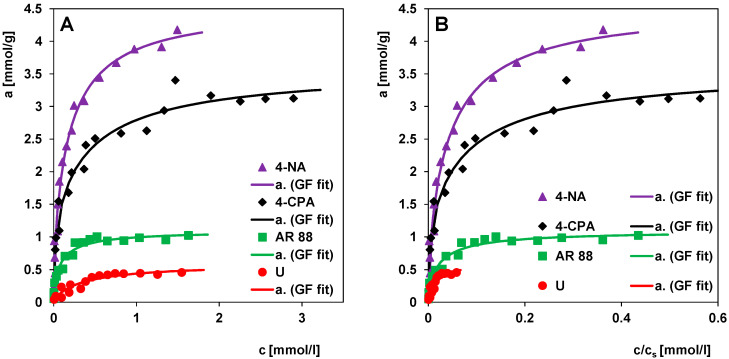
Comparison of adsorption isotherms of various pollutants on the activated carbon RIAA in the linear coordinate system (**A**) and in the reduced linear coordinate system (**B**).

**Figure 8 molecules-30-00088-f008:**
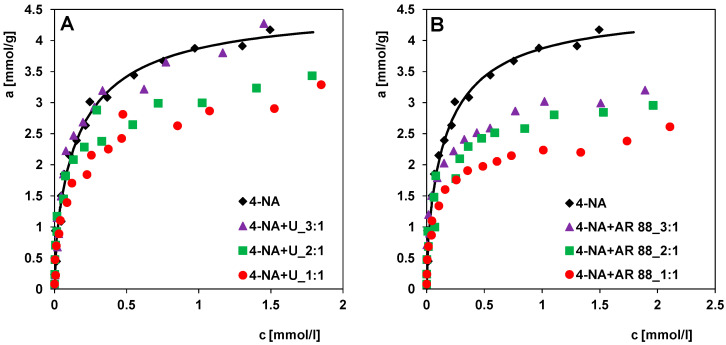
Adsorption isotherms of (**A**) 4-nitroaniline in the presence of uranine (4-NA + U), (**B**) 4-nitroaniline in the presence of Acid Red 88 (4-NA + AR 88) compared to one-component system (4-NA). The concentration ratios of 4-NA to dye in the initial solutions were 3:1, 2:1, 1:1.

**Figure 9 molecules-30-00088-f009:**
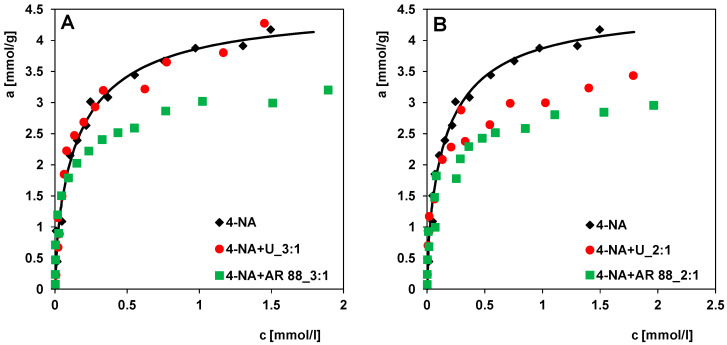
Adsorption isotherms of 4-nitroaniline in the presence of uranine (4-NA + U) and 4-nitroaniline in the presence of Acid Red 88 (4-NA + AR 88) at the concentration ratios of 4-NA to dye in the initial solutions of 3:1 (**A**), 2:1 (**B**), and 1:1 (**C**) compared to one-component system (4-NA). (**D**) Adsorption isotherms of Acid Red 88 in the presence of 4-nitroaniline (AR 88 + 4-NA) compared to one-component system (AR 88). Comparison of the adsorption isotherms of 4-nitroaniline (4-NA).

**Figure 10 molecules-30-00088-f010:**
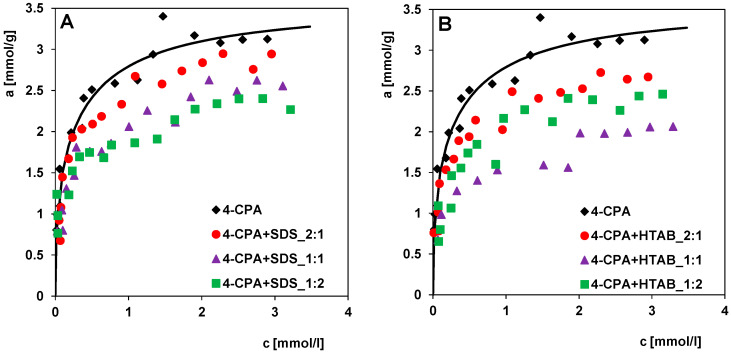
Adsorption isotherms of (**A**) 4-chlorophenoxyacetic acid in the presence of sodium dodecyl sulfate (4-CPA + SDS), and (**B**) 4-chlorophenoxyacetic acid in the presence of hexadecyltrimethylammonium bromide (4-CPA + HTAB) compared to one-component system (4-CPA). The concentration ratios of 4-CPA to surfactant in the initial solutions were 2:1, 1:1, 1:2.

**Figure 11 molecules-30-00088-f011:**
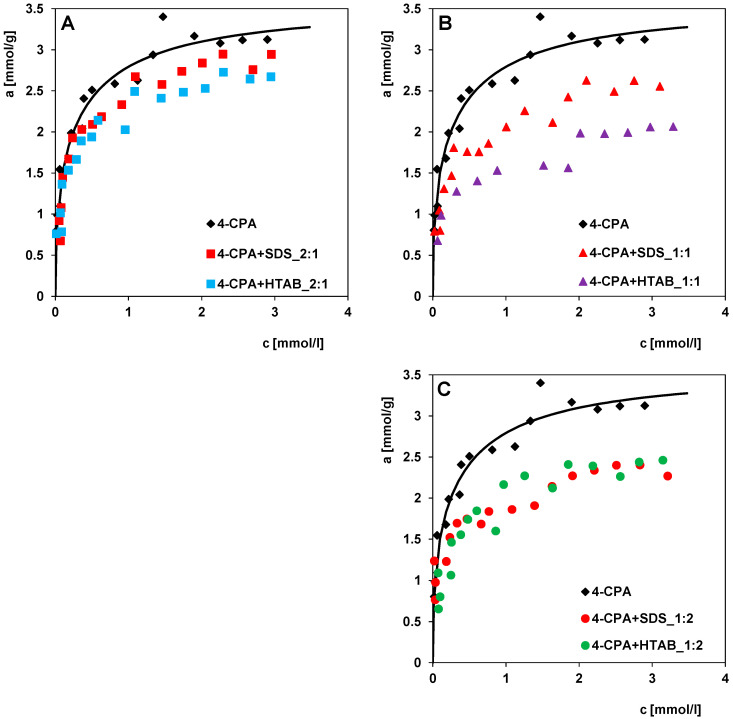
Adsorption isotherms of 4-chlorophenoxyacetic acid in the presence of sodium dodecyl sulfate (4-CPA + SDS) and 4-chlorophenoxyacetic acid in the presence of hexadecyltrimethylammonium bromide (4-CPA + HTAB) at the concentration ratios of 4-CPA to surfactant in the initial solutions of 2:1 (**A**), 1:1 (**B**), 1:2 (**C**) compared to one-component system (4CPA).

**Table 1 molecules-30-00088-t001:** The values of parameters that characterise the porous structure of adsorbents.

Carbon	S_BET_ ^a^ [m^2^/g]	V_t_ ^b^ [cm^3^/g]	V_mic_ ^c^ (t-Plot) [cm^3^/g]	D_h_ ^d^ [nm]	D_mo_ ^e^ (des. BJH) [nm]	D_mo_ ^f^ (HK) [nm]	D_mo_ ^g^ (DFT) [nm]	pH_pzc_ ^h^
RIAA	1468	0.80	0.31	2.2	2.56	0.75	0.66	9.2

^a^ BET surface area calculated using experimental points at a relative pressure of (p/p_0_) 0.035–0.31, where p and p_0_ are denoted as the equilibrium and saturation pressure of nitrogen. ^b^ Total pore volume calculated by 0.0015468 amount of nitrogen adsorbed at p/p_0_ = 0.98. ^c^ Pore volume of micropores calculated by t-plot method with fitted statistical thickness in the range of 3.56 to 4.86 Å. ^d^ The average pore diameter estimated based on the BJH theory. ^e^ The average pore diameter is estimated based on the BJH theory from desorption data. ^f^ Micropore size distributions determined as the Horvath–Kawazoe differential pore volume plot and ^g^ Nonlocal density functional theory (NLDFT). ^h^ The pH at the point of zero charge.

**Table 2 molecules-30-00088-t002:** The parameters of the Generalised Freundlich equation (GF) for organic substances’ adsorption on the activated carbon RIAA.

Adsorbate	a_m_ [mmol/g]	m	n	log K [L/mmol]	R^2^
4-NA	4.55	0.55	1	0.47	0.98
4-CPA	3.59	0.36	1	−0.01	0.96
AR 88	1.09	0.47	1	0.65	0.97
U	0.59	0.74	1	0.29	0.89

**Table 3 molecules-30-00088-t003:** Comparison of the adsorption efficiency of the discussed compounds on various types of sorbents.

Adsorbate	Adsorbent Type	Adsorption mmol/g	References
4-NA	carbon fibre from cotton stalk	2.94	[56]
	MCM-48	0.65	[61]
	carboxylated polymeric adsorbent	2.41	[62]
	activated carbon RIAA	4.55	this work
4-CPA	granular activated carbon F-400	1.47	[57]
	activated carbon	2.26	[58]
	activated carbon after abrasion	1.73	[58]
	activated carbon RIAA	3.59	this work
AR 88	activated carbon from cypress sawdust	0.805	[59]
	mesoporous activated carbon	0.772	[63]
	bentonite	0.23	[64]
	activated carbon RIAA	1.09	this work
U	activated carbon	0.05	[60]
	chitosan	5.4 × 10^−4^	[65]
	metal-organic framework	0.38	[60]
	activated carbon RIAA	0.59	this work

## Data Availability

The data presented in this study are available on request from the corresponding author.

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
