# Peer review of "Equilibrium Adsorption of Organic Compounds (Nitrobenzene Derivative, Synthetic Pesticide, Dye, and Surfactant) on Activated Carbon from Single- and Multi-Component Systems"

_molecules, 2024, doi:10.3390/molecules30010088_

Round 1

Reviewer 1 Report

Comments and Suggestions for Authors

Equilibrium adsorption of organic pollutants on activated car3 bon from single- and multi-component systems

The authors make a well characterization of one commercial activated carbon. Then tested in the adsorption of contaminant in the presence of cosurfactant. This is quite important in the water depuration process.

Just some minor comments are:

In the abstract please indicate what is activated carbon RIAA, it is a commercial one? form what company?

Figure 3. XPS. In the y axis, there is no need to indicated the number if the unit are arbitrary units.

Figure 4. Can be include FTIR analysis to corroborate the surface groups?

Perhaps Table 3 and Figure 8 could be in SI if there is need to shorten the paper.

There is any possibility to calculi the difference between the adsorption capacity of the contaminant and the contaminant plus surfactant, and represent this difference vs activated carbon properties, just to look for some correlations.

It is possible to see a direct correlation between the pollutant size and the micro and meso porosity amount?

Check if this adsorption competition is kept in real depurated water would give the paper a very good application

Reviewer 2 Report

Comments and Suggestions for Authors

The paper presents theoretical and experimental information regarding the equilibrium adsorption of 4-nitroaniline and 4-chlorophenoxyacetic acid on activated carbon in the presence of organic dyes and surfactants. The data obtained are important in wastewater treatment processes.

Although the presented study is comprehensive and the presentation is mostly clear, some important aspects should be improved.

-        The title of the paper is much too general. I suggest the class of organic pollutants or namely the main contaminants that are considered in the study to be specified. In the submitted form of the manuscript, 4-nitroaniline and 4-chlorophenoxyacetic acid are not the only organic contaminants from the simulated aqueous solutions. Also, the specificity of the presented study cannot be safely generalized.

-        The introduction should be restructured, preserving aspects specific to the organic compounds that are the subject of the actual research. No comparisons of the experimental data from this research with data from the literature are presented. In relation to the introductory paragraphs, a final comparative table should be considered in the section of Results and Discussion in order position the results among those reported in literature. Without this information it is difficult to estimate whether the obtained results have any practical importance.

-        References related to the separation of 4-nitroaniline and 4-chlorophenoxyacetic acid from water should be introduced or it should be specified that the paper presents for the first time a method for the separation of 4-nitroaniline and 4-chlorophenoxyacetic acid on activated carbon.

-        Line 218 – explain “a”.

-        Line 282 -reformulate “thesis”.

-        In the abstract section it is mentioned the use of GL isotherm model, whereas in in the section of Results and Discussion, it seems data were modelled with GF model. Also, eqs. (3)-(7) are not relevant for the presented study. The authors should present only the model they applied, which should be discussed in the abstract, results and conclusions.

-        The meaning of the solid line should be explained in Figs 9-12. If it is obtained based on a model, what are the parameters of the model? This description must be added also in figures captions.

-        The conclusions presented are incomplete. General specifications are presented such as: decreases, increases, higher, lower. No actual improved or diminished results are specified for adsorption in a multicomponent system. What is the effective influence of a component on the retention of 4-nitro aniline and 4-chlorophenoxyacetic acid. Overall, the conclusions seem to present data that have not been presented. The authors should to refer only to the presented data.

Reviewer 3 Report

Comments and Suggestions for Authors

The topic of the manuscript is really interesting and well-rounded, but I have some concerns about the results presentation, specially with two adsorbate systems.

The regression analysis made in simple solutions have no confidence interval of each parameter of the isotherm model used in this work. Also, it is highly recommended to show the units for K. 

There are no statistical/experimental proof of the qualitative results and discussions for two adsorbate systems. It is needed to assess significant differences among every comparison.

Also, it is suggested to use the same regression analysis that were used in point 2.2 on two adsorbate mixtures. 

Round 2

Reviewer 2 Report

Comments and Suggestions for Authors

The authors have addressed most of the issues. However, there are a few confusing information. The abstract section still refers to Generalized Langmuir model. Also, the caption of the several figures mention that the solid lines correspond to trendlines. The specific used isotherm model should be mentioned.

Reviewer 3 Report

Comments and Suggestions for Authors

Experimental/statistical proof was not delivered as requested.  
